# Increased Sensitivity of PBMCs Isolated from Patients with Rheumatoid Arthritis to DNA Damaging Agents Is Connected with Inefficient DNA Repair

**DOI:** 10.3390/jcm9040988

**Published:** 2020-04-01

**Authors:** Grzegorz Galita, Olga Brzezińska, Izabela Gulbas, Joanna Sarnik, Marta Poplawska, Joanna Makowska, Tomasz Poplawski

**Affiliations:** 1Department of Molecular Genetics, Faculty of Biology and Environmental Protection, University of Lodz, 90-236 Lodz, Poland; 2Department of Rheumatology, Medical University of Lodz, 92-115 Lodz, Poland; 3Biobank, Department of Immunology and Allergy, Medical University of Lodz, 92-213 Lodz, Poland

**Keywords:** DNA damage and repair, rheumatoid arthritis, oxidative DNA lesions, DNA double-strand breaks, comet assay

## Abstract

Rheumatoid arthritis (RA) is a systemic, inflammatory disease of the joints and surrounding tissues. RA manifests itself with severe joint pain, articular inflammation, and oxidative stress. RA is associated with certain types of cancer. We have assumed that RA patients’ increased susceptibility to cancer may be linked with genomic instability induced by impaired DNA repair and sensitivity to DNA damaging agents. The aim of this work was to analyze the sensitivity of peripheral blood mononuclear cells (PBMCs) isolated from RA patients to DNA damaging agents: tert-butyl hydroperoxide (TBH), bleomycin, ultraviolet (UV) radiation, and methyl methanesulfonate (MMS) and calculate the repair efficiency. TBH induce oxidative DNA lesions repaired mainly by base excision repair (BER). Bleomycin induced mainly DNA double-strand breaks repaired by non-homologous end joining (NHEJ) and homologous recombination repair (HRR). We included 20 rheumatoid arthritis patients and 20 healthy controls and used an alkaline version of the comet assay with modification to measure sensitivity to DNA damaging agents and DNA repair efficiency. We found an increased number of DNA breaks and alkali-labile sites in the RA patients compared to those in the controls. Exposure to DNA damaging agents evoked the same increased damage in both groups, but we observed statistically higher PMBC sensitivity to TBH, MMS, bleomycin as well as UV. Examination of the repair kinetics of both groups revealed that the DNA lesions induced by TBH and bleomycin were more efficiently repaired in the controls than in the patients. These data suggest impaired DNA repair in RA patients, which may accelerate PBMC aging and/or lead to higher cancer incidence among RA patients.

## 1. Introduction

Rheumatoid arthritis (RA) is the most common chronic, autoimmune arthritis affecting 1% of people worldwide. RA affects not only joints and the musculoskeletal system but also leads to systemic uncontrolled inflammation resulting in internal organ damage and decreased life expectancy. Moreover it is known that patients with RA are at high risk of developing certain neoplasmatic disorders, especially diffuse large B-cell lymphoma (DLBCL) [1]. The question this raises, however, is whether this susceptibility to cancers is the result of the disease itself, of exposure to chronic inflammation and persistent oxidative stress, or a result of the drugs used to treat patients. We hypothesized that RA patients’ susceptibility to cancers is a consequence of unrepaired DNA damage. In response to DNA damage, the cell triggers the DNA damage response (DDR). DDR is a multistep process that includes DNA lesions recognition, protein recruitment, DNA lesions removal, and reconstruction of DNA structure. Any deregulation in the activity of DNA repair systems resulting in genomic instability and may be linked with an increased susceptibility to cancer [1]. We presume that RA patient’s increased susceptibility to cancer may be associated with genomic instability in terms of DNA damage and repair and susceptibility to DNA damaging agents. This hypothesis is supported by the growing list of reports that suggest altered DNA repair in RA patients [2,3,4,5,6]. However, these reports suffer from inadequate selection of the study group and superficial analysis. Moreover, the authors focused only on signs of impaired DRR such as telomere length, presence of DNA strand breaks, or expression of DDR control proteins. The aim of our study was to evaluate the level of endogenous damage and sensitivity of peripheral blood mononuclear cells (PBMCs) isolated from RA patients to various DNA damaging agents. We also analyzed the efficiency of DNA repair.

## 2. Experimental Section

The study group included 20 patients with rheumatoid arthritis (16 women and 4 men; mean age 58.85 ± 15.15 years) selected from patients of the Department of Rheumatology and outpatient clinic. This cohort study has been approved by the Institutional Bioethics Committee of the Medical University of Lodz (Lodz, Poland) (no. RNN/07/18/KE, approved date: 16 January 2018). All patients fulfilled the EULAR/ACR 2010 diagnostic criteria of rheumatoid arthritis. Mean time of disease duration was 10.15 ± 10.96 years (from 1 to 38 years). Ten patients were currently (for at least one month before blood collection) treated with methotrexate, three patients with sulfasalazine, and seven patients did not receive disease modifying anti-rheumatic drugs (DMARDs) within the last month. Seven patients were taking glucocorticosteroids (GCS) (within the last week). All patients had rheumatoid factor levels (positive in 16 cases) and the presence of anti-citrullinated protein antibodies (positive in 17 cases). In addition, the level of inflammation markers (ESR- erythrocyte sedimentation rate) 24.58 ± 15.85 mm/h; C-reactive protein, CRP 17.78 ± 23.80 g/dL) was determined (more details in Table 1). The disease activity has been also assessed based on Disease Activity Score 28-joint count C reactive protein (DAS28) -CRP score (DAS <1.7 was defined as remission, DAS >1.7 and <2.6 was defined as low disease activity and DAS28 above 5.1 as high disease activity). A control group of 20 volunteers (16 women and 4 men; mean age 63.5 ± 9.85) was recruited from patients without symptoms of chronic inflammatory conditions. All controls had ESR (erythrocyte sedimentation rate) and CRP (C-reactive protein) within normal limits and did not have any chronic disease of inflammatory background. The exclusion criteria for both the study and control groups were past or presence malignancy as a potential reason for DNA instability.

We determined PBMC sensitivity to DNA damaging agents as well as DNA repair efficiency. Analysis of the effectiveness of DNA repair and susceptibility to DNA damaging agents was carried out using the alkaline version of the comet assay with modifications [7,8]. The DNA damaging agents selected for study induce various DNA lesions, which are subjected to two excision DNA repair pathways: base excision repair (BER) and nucleotide excision repair (NER) and two DNA double-strand break (DSB) repair pathways: non-homologous end joining (NHEJ) and homologous recombination repair (HRR). Tert-butyl hydroperoxide (TBH) induces oxidative DNA lesions and these types of lesions are repaired mainly by the BER pathway. Bleomycin mimics the ionizing radiation effects and induces mainly DSB repairs, which are by NHEJ and HRR. We also used ultraviolet (UV) radiation as a DNA damaging agent. It generates the formation of pyrimidine dimers in DNA, which are substrates for NER. These types of DNA lesions cannot be measured directly by the comet assay, therefore we utilized in the comet assay the T4 PDG (T4 pyrimidine DNA glycosylase) enzyme. T4 PDG “breaks” the glycosyl bond at the 5′ end of the pyrimidine dimer and the phosphodiester bond at the AP (apurinic/apyrimidinic site) site. As a result, DNA strand breaks were generated, which can be evaluated by the comet assay. Similar to UV radiation, methyl methanesulfonate (MMS) introduced methyl adducts in DNA (repaired mainly by BER [9]) that also cannot be measured directly by the comet assay. We used aphidicolin to break the repair process at the final stage of BER. This results in DNA strand breaks, which can be evaluated by the comet assay. All compounds in the concentrations used for study did not affect the viability of PMBCs (viability of PMBCs was always >80%). We used TBH at 7 µM for 15 min on ice, bleomycin at 25 µM for 30 min at 37 °C, MMS at 3 µM, and aphidicolin at 15 µM for 120 min at 37 °C.

The data are presented as the median ± range. The Mann–Whitney rank sum test was used to compare DNA damage between patients with RA and healthy controls. The normal distribution of continuous variables was confirmed by the Shapiro–Wilk test. Mann–Whitney rank sum test was decided based on the normality test. ANOVA on ranks was used to compare many groups followed by all pairwise multiple comparison procedures (Dunn’s method). Receiver Operating Characteristic (ROC) curves analysis was conducted to compare DNA repair curves between RA patient and healthy controls. In all tests, p value <0.05 was used.

All statistical analyses were performed with TIBCO Statistica 13.3 (Palo Alto, CA 94304, USA).

## 3. Results

Analysis of the level of endogenous damage with Mann–Whitney rank sum test (Figure 1A) showed, increased statistically significant level (*p* < 0.001) in PBMCs isolated from RA patients compared to healthy patients median RA = 9.280 (25% = 4.010; 75% = 17.540) vs. 2.090 in control (25% = 0.750; 75% = 4.825). The extent of the DNA damage induced by TBH (Figure 1B) (22.330; 25% = 12.448, 75% = 35.560 vs. 9.325; 25% = 3.732; 75% = 18.732) as well as bleomycin (Figure 1C) (44.190; 25% = 28.385, 75% = 61.263 vs. 12.370; 25% = 4.925, 75% = 26.050) was significantly higher in PBMCs derived from RA patients than in those from healthy counterparts (*p* < 0.001). Similarly, an increased, statistically significant level of DNA damage in PBMCs isolated from RA patients was observed after exposure to MMS (Figure 1D) (median RA = 22.730, 25% = 12.468, 75% = 38.640 vs. 8.330, 25% = 3.615, 75% = 16.410 in controls). However, we did not notice a difference in the level of DNA damage between PBMCs exposed to UV radiation isolated from RA patients and compared to control (Figure 1E) (median RA = 44.970, 25% = 30.650, 75% = 61.050 vs. 39.890; 25% = 24.255, 75% = 56.220 in controls). After taking into account the baseline damage, the following values were obtained: TBH (Figure 1B’) (13.280; 25% = 3.180, 75% = 26.050 vs. 6.615, 25% = 2.00, 75% = 16.130), bleomycin (Figure 1C’) (34.650; 25% = 18.615, 75% = 52.680 vs. 9.890; 25% = 2.900, 75% = 23.165), methyl methanesulfonate (Figure 1D’) (13.670; 25% = 3.273, 75% = 29.190 vs. 5.900; 25% = 2.90, 75% = 13.445), and UV radiation (Figure 1E’) (36.095; 25% = 21.423, 75% = 52.227 vs. 37.610; 25% = 22.285, 75% = 54.080).

We also analyzed the efficiency of DNA repair in PBMCs isolated from RA patients by estimating the level of damage induced by DNA damage compounds during the 120 min of repair process compared to healthy control. The kinetics of DNA repair from RA patients (dashed line) after the introduction both bleomycin or TBH was statistically different as compared to that of the healthy controls (solid line)— ROC area curve 0.8105; *p* < 0.001 for TBH (Figure 2A) and ROC area curve 0.7732; *p* < 0.001 for bleomycin (Figure 2B).

As disease treatment, exacerbation, or concomitant syndromes can affect DNA damage we performed a clinical sub-analysis (Figure 3). ANOVA on ranks with multiple comparisons versus control group (Dunn’s method) showed that PBMCs isolated from RA patients without or with treatment are characterized by a much higher level of endogenous DNA damage (*p* < 0.001). Treatment with methotrexate but not glucocorticosteroids is connected with the increasing level of endogenous DNA damage as compared with RA patients without treatment (Figure 3A, median 10.860, 25% = 5.34, 75% = 19.37 vs. 7.24, 25% = 2.67, 75% = 15.54 and 7.070, 25% = 3.10, 75% = 13.04).

Patients with high disease activity (defined as DAS28 higher than 5.1) show similar level of endogenous damage as patients with low disease activity or remission (defined as DAS28 < 2.6) (Figure 3B; median 9.85, 25% = 4.03, 75% = 19.87 vs. 8.05, 25% = 2.65, 75% = 17.28). Anyway, these two groups showed more endogenous damage than the control group (multiple comparisons versus control group (Dunn’s method), (*p* < 0.001).

One of our patients had secondary Sjogren syndrome concomitant to RA, we also found that one of the patients presented Felty syndrome, which is a rare complication of RA but connected to an increased risk of lymphoma. Therefore, we compared the level of endogenous DNA damage in PMBCs isolated from these patients. The patients with Felty syndrome had the highest level of DNA damage and the one with Sjogren syndrome had the lowest (median 14.76, 25% = 9.22, 75% = 22.25 vs. 5.48, 25% = 3.507, 75% = 8.42). However, we did not perform statistical analysis due to the very small group number of the patients (*n* = 1).

## 4. Discussion

According to our observations, PBMCs isolated from RA patients are characterized by greater sensitivity to DNA damaging factors as compared to those from healthy subjects. The obtained results suggest that this phenomenon is related to delayed oxidative DNA lesion repair and sensitivity to alkylating agents, and it suggest that BER is affected in RA. We estimated the level of damage and the effectiveness of PBMC repair for tert-butyl hydroperoxide (TBH; T-BOOH). TBH induces oxidative stress in eukaryotic cells [10]. TBH significantly increased the amount of 8-oxoguanine (8-oxoG) in DNA. 8-oxo-G is the main substrate for glycosylases involved in BER. Therefore, the increased level of PBMC sensitivity to damage induced by TBH and less effective repair relative to the control group may suggest a lower efficiency of the BER system in RA patients than in healthy subjects. The emerging oxidative stress can have many causes, including NADPH oxidase, accumulation of reactive oxygen species and their direct and indirect interaction in the cell, and insufficient amount of antioxidants. Oxidative stress can also contribute to increased sensitivity to DNA oxidative damage, and those not repaired by an inefficient repair system can lead to cell death [11]. Elevated sensitivity of PBMCs isolated from RA patients to DNA damaging agents that trigger BER suggests that one key component of BER is affected, and an additional study with a functional BER assay and Western blot analysis is needed to answer the question “Which component of BER does not function well in RA”. Moreover, since RA manifests itself with articular inflammation and oxidative stress, an effective BER system that deals with oxidative DNA lesions is a key element in ensuring the genomic stability of RA human cells. We also showed that DSBs repair is delayed in RA patients as compared with healthy subjects during 120 min, and we have seen greater sensitivity in PBMCs isolated from patients to bleomycin. Similar observations regarding delayed DSBs repair in RA were published earlier, however DSBs repair was performed on very small group. It consisted of seven patients belonged to one, different than our, ethnic group (African Americans) [12]. Since racial and ethnic disparities exist for minority populations with RA, the conclusions of this study should be treated with caution. Our study also has limitations. It had a small sample size, which also limited subgroup comparisons (newly diagnosed RA patients vs. treated, chronic vs. acute, preclinical and early phases of RA development vs. late phase of RA). We included RA patients under RA therapy. This could have affected observations as many RA drugs (especially MTX) induce DNA breaks. This explains the higher level of DNA strand breaks observed in RA patients as compared with healthy subjects by us and others [2,3,12], but it does not explain impaired oxidative DNA lesions as well as DSBs repair. We also noticed that GCS treatment is much better for patients in terms of DNA damage presented in PMBCs. Higher endogenous DNA level in PMBCs isolated from RA patients is characteristic to RA or common for autoimmune disorders prone to malignancies such as Sjogren syndrome and dermato/polymyositis. We suggest that this is rather RA specific as we showed no difference between patients with inflammatory idiopathic myopathies and the control group of healthy volunteers [13]. We performed a sub-analysis of RA patients comparing patients with high and low disease activity. We did not find differences; however, the subgroups were relatively small to enable a final conclusion, and further studies are required to address the question.

## 5. Conclusions

In conclusion, impaired DNA repair in RA patients may accelerate PBMC aging and contribute to RA phenotype. The question of whether the defect in DNA repair is a consequence of RA or one of RA susceptibility markers is still unanswered. The next question concerns the mechanism of inefficient DNA repair in RA. Functional, genetic, or epigenetic factors may contribute to this phenomenon and they need to be identified.

## Figures and Tables

**Figure 1 jcm-09-00988-f001:**
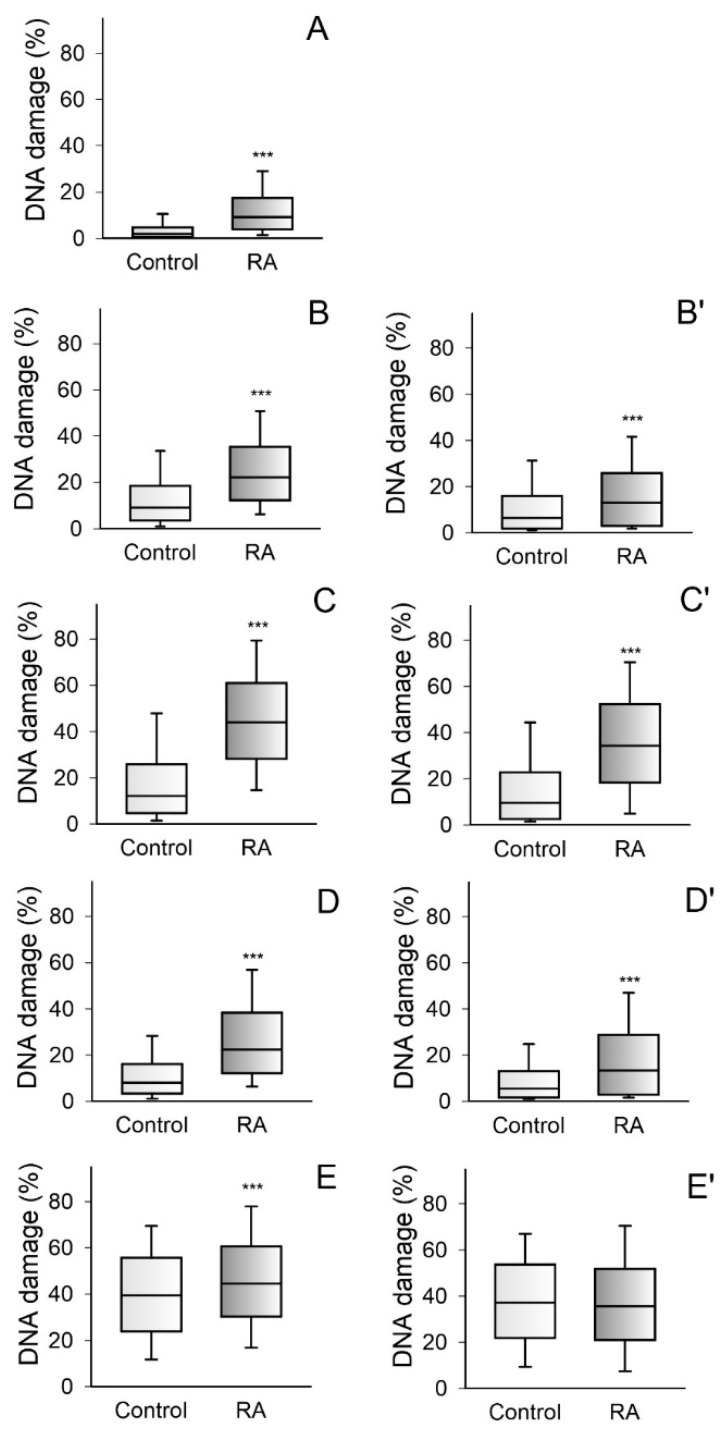
Peripheral blood mononuclear cells (PBMCs) isolated from rheumatoid arthritis (RA) patients were more sensitive to DNA damaging agents than PBMCs isolated from healthy subjects. Endogenous DNA damage (**A**), DNA damage resulting from exposure to DNA damaging agents: tert-butyl hydroperoxide (7 µM) before (**B**) and after consideration of the baseline damage (**B’**), bleomycin (25 µM) (**C**,**C’**), methyl methanesulfonate (3 µM) (**D**,**D’**), and UV radiation (**E**,**E’**) in PBMCs of 20 healthy controls and 20 RA patients. DNA damage was measured as the percentage of DNA in the tail in the alkaline version of the comet assay. The value of cells scored for each individual was 100. The sensitivity of DNA damage was calculated immediately after incubation with drugs. Differences between groups were analyzed using the Mann–Whitney rank sum test Analysis, *** means *p* < 0.001.

**Figure 2 jcm-09-00988-f002:**
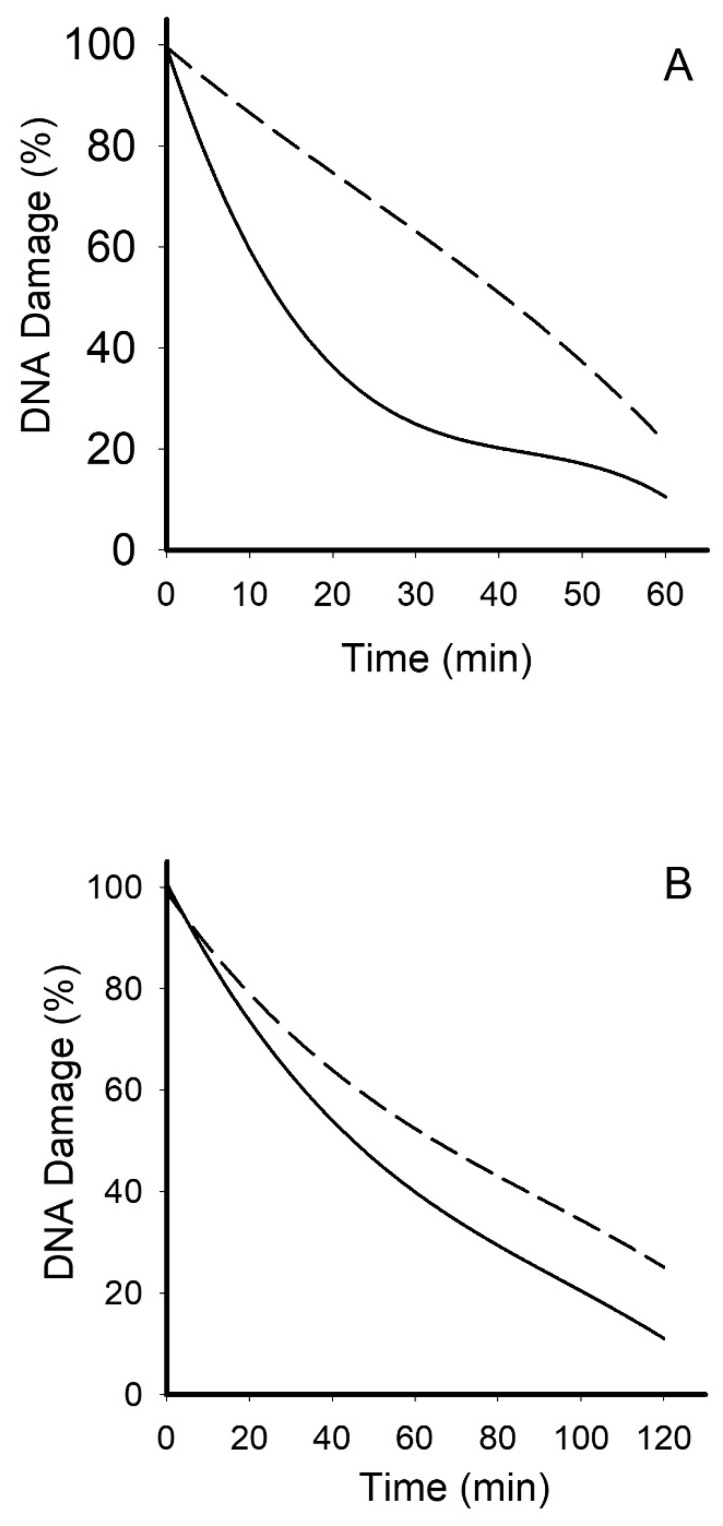
Peripheral blood mononuclear cells (PBMCs) isolated from rheumatoid arthritis (RA) patients have less efficient DNA repair than PBMCs isolated from healthy subjects. Repair of DNA lesions evoked by tert-butyl hydroperoxide (**A**) or bleomycin (**B**) in the PBMCs of 20 healthy control (solid line) and 20 RA patients dashed line). PBMCs were allowed to recover their DNA for 60 min after incubation with tert-butyl hydroperoxide and 120 min after incubation with bleomycin. DNA damage was measured as the percentage of DNA in the tail in the alkaline version of the comet assay. The value of cells scored for each individual was 100. Curves were compared using Receiver Operating Characteristic (ROC) curve.

**Figure 3 jcm-09-00988-f003:**
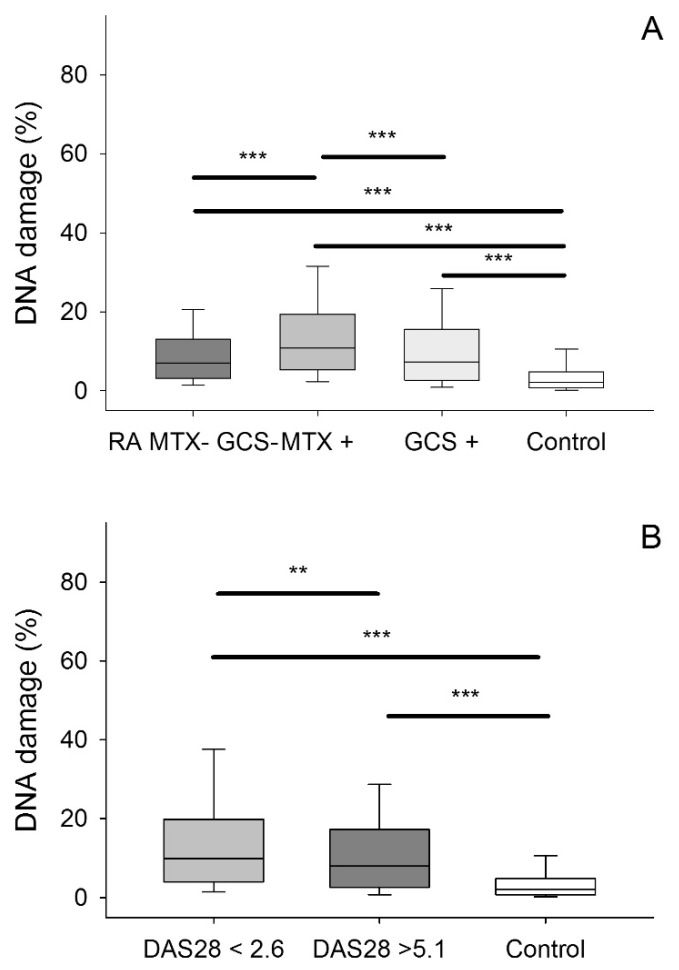
Subclinical analysis of the endogenous DNA lesion level in peripheral blood mononuclear cells (PBMCs) isolated from rheumatoid arthritis (RA) patients. (**A**) Treatment with disease modifying anti-rheumatic drugs such as methotrexate (MTX+, dark grey, *n* = 10) as well as glucocorticosteroids (GCS +, light grey, *n* = 7) increased the basal endogenous DNA damage level in comparison to RA without treatment (MTX−, GCS−, grey, *n* = 2). However, the basal endogenous DNA damage level in PMBCs isolated from non-treatment patients was greater than the basal endogenous DNA damage level in PMBCs isolated from healthy subjects (Control, white, *n* = 20). (**B**) The RA exacerbation had no impact on basal endogenous DNA damage level in PMBCs isolated from RA patients. The RA exacerbation was presented by the Disease Activity Score 28-joint count (DAS28) parameter. Patients with high disease activity were defined as DAS28 higher than 5.1 (dark grey, *n* = 6); whereas patients with low disease activity or remission were defined as DAS28 < 2.6 (grey, *n* = 5). White box means control. DNA damage was measured as the percentage of DNA in the tail in the alkaline version of the comet assay. The value of cells scored for each individual was 100. The differences between two groups were analyzed using the Mann–Whitney rank sum test. Differences between more than two groups were analyzed using an ANOVA on ranks test following all pairwise multiple comparison procedures (Dunn’s method). Analysis, *** means *p* < 0.001, ** means *p* < 0.01.

**Table 1 jcm-09-00988-t001:** Clinical characteristics of the studied group. CRP (C-reactive protein), ESR (erythrocyte sedimentation rate), RF (rheumatoid factor), ACPA (anti-citrullinated peptide antibodies), GCS (glucocorticosteroids), F-Female, M-Male.

Title 1	RA patients *n* = 20	Controls *n* = 20
Sex	F 16; M 4	F 16; M 4
Age	58.85 ± 15.15	63.5 ± 9.85
Disease Duration	10.15 ± 10.96	
Remission	yes 5; no 15	
CRP	17.78 ± 23.80	
ESR	24.58 ± 15.85	
RF	202.16 ± 265.82	
ACPA	1934.12 ± 5577.19	
Treatment	Methotrexate 10/20Sulfasalazine 3/20Glucocorticosteroids 7/20

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
