# Peer review of "Increased Sensitivity of PBMCs Isolated from Patients with Rheumatoid Arthritis to DNA Damaging Agents Is Connected with Inefficient DNA Repair"

_jcm, 2020, doi:10.3390/jcm9040988_

Round 1

Reviewer 1 Report

In the present manuscript the Authors tempted to demonstrate the higher incidence of DNA damage and low DNA repair activity secondary to RA in a small cohort of 20 adults. Do the laboratory methodology be affected by the disease exacerbation? In that case, I would suggest to confirm the results in a cohort of patients after the resolution of the disease flare. Did the members of the control group be screened for potential subclinical inflammatory conditions? Did they presented any other concomitant diseases? Please report additive data about the control group in table 1. Minor suggestion: please spelling DLBCL at line 48, page 2.

Author Response

Point 1: In the present manuscript the Authors tempted to demonstrate the higher incidence of DNA damage and low DNA repair activity secondary to RA in a small cohort of 20 adults. Do the laboratory methodology be affected by the disease exacerbation? In that case, I would suggest to confirm the results in a cohort of patients after the resolution of the disease flare.

Response 1: To rule out the effect of disease activity we analyzed patients with high disease activity (defined as DAS28 higher than 5.1) with patients with low disease activity or remission (defined as DAS28 < 2.6) (see methodology and result section and figure 3). We have not found difference.

Point 2: Did the members of the control group be screened for potential subclinical inflammatory conditions? Did they presented any other concomitant diseases? Please report additive data about the control group in table 1.

Response 2: All controls had ESR and CRP within normal limits and did not have any chronic disease of inflammatory background, which in our opinion could probably exclude subclinical systemic inflammation. We have added these information into manuscript

Point 3: Minor suggestion: please spelling DLBCL at line 48, page 2.

Response 3: We have done so.

With many thanks for valuable comments,

Tomasz Poplawski

Reviewer 2 Report

The paper is interesting and covers an aspect that is worthy.

The authors in their limitation section indicate that the effects might be caused by the concurrent therapy with methotrexate and leflunomide. I believe this issue is of critical importance, therefore they should also present the data of RA patients segregated into two; those treated with the above mentioned medications and those not, since the effects described may be entirely related to the medications’ effect and the disease’s.

It would be of interest if the authors discussed similar reports if exist in disorders that are more prone to malignancies such as Sjogren syndrome and dermato/polymyositis.

Author Response

Response to Reviewer 2 Comments

Point 1: The authors in their limitation section indicate that the effects might be caused by the concurrent therapy with methotrexate and leflunomide. I believe this issue is of critical importance, therefore they should also present the data of RA patients segregated into two; those treated with the above mentioned medications and those not, since the effects described may be entirely related to the medications’ effect and the disease’s.

Response 1: We agree that the influence of the drugs taken by the patient cannot be neglected.  Unfortunately due to ethical issues we couldn’t stop the treatment in our patients. We are planning to collect the group of newly diagnosed RA patients (disease modifying anti-rheumatic drugs, DMARDS, naive patients). We analyzed one more time the current taken drugs and history of the drug taken. We defined current treatment with DMARDs as drugs taken within last month before blood collection. We defined current treatment with glucocorticosteroids as GCS treatment within last week before blood collection. To answer the question of the Reviewer we sub-analyzed our patients dividing them into the subgroup treated with MTX within last month as compared to DMARDs naive patients.  We also compared subjects receiving GCS to GCS naive patients. The analysis and figures were added to result section.

Point 2: It would be of interest if the authors discussed similar reports if exist in disorders that are more prone to malignancies such as Sjogren syndrome and dermato/polymyositis.

Response 2: In the matter of fact we have some data on patients with inflammatory idiopathic myopathies which showed no difference between studied group and control group of healthy volunteers (These results were presented during EULAR 2018). We did not continue to study those patients as preliminary data did not show any promising results. We mentioned patients with inflammatory idiopathic myopathies in the discussion. However, we think that inflammatory idiopathic myopathies are in the matter of fact group of heterogenous entities and as far we know not all subtypes or autoantibodies are connected with increased risk of neoplasmatic disorders. To our preliminary data we included patients with mainly antisynthethase syndrome which do not closely correlate with cancers , maybe that’s why we did not observed any significant difference. We agree that patients with Sjogren syndrome would be of great interest as those patients have even 40 times higher risk to develop lymphoma. So far we have little data on Sjogren syndrome and DNA stability. However interestingly one of our patients had secondary Sjogren syndrome concomitant to RA, we also  found that one of the patients presented Felty syndrome which is a rare complication of RA but connected to increased risk of lymphoma. We presented those patients separately in Results section.

Reviewer 3 Report

In the manuscript by Galita et al., the authors assume that increased susceptibility to cancer of RA patients may be linked with genomic instability induced by impaired DNA repair and sensitivity to DNA damaging agents.

The authors analyze the sensitivity of peripheral blood mononuclear cells (PBMC) isolated from 20 rheumatoid arthritis patients to DNA damaging agents: tert-Butyl hydroperoxide (TBH), bleomycin, UV radiation and MMS compared to 20 healthy controls. Sixteen of them was treated by disease modification drugs (methotrexate 55%; sulfasalazine 20%;leflunomide 5%; hydroxychloroquine 5%).

The authors utilize a comet assay and a modified comet assay T4 PDG enzyme. The latter generates DNA strand breaks, which can be evaluated by comet assay.

The Increased sensitivity to DNA-damaging agents of RA patients can be linked to impaired DNA repair is of certain relevance. However, the authors do not provide some informations (drug concentration and timing of treatment) and do not show convincing data supporting their hypothesis.

Main issues:
Results
Figure 1A shows high baseline levels of endogenous damage in RA patients compared to control (median RA 9.280 vs 2.090 in control, ratio 4.44). This is expected since RA patients take drugs that can cause DNA damage or inhibit DNA repair. In fact, Ra patients are 4.44 times more damaged than controls. The same authors in the conclusions state that the different basal levels of damage may be dependent on the drugs taken by the RA patients (lines 155 -158).

Figures 1B-1D show the percent of damage induced by TBH, bleomycin, methyl methanesulfonate and UV radiation between RA patients and control.
1) How can the authors state that the observed differences are significant considering baseline differences between the two groups?
2) Have the authors measured drug treatment-induced cell death levels?
3) Have the authors considered that the ratio between the two groups tends to drop after the various treatments?
see baseline levels(Fig.1A, median RA 9.280 vs 2.090 in control, ratio 4.44)
TBH (Fig.1B, median RA 22.330 vs 9.325 in control, ratio 2.39)
bleomycin (Fig.1C, median RA 44.190 vs 12.370 in control, ratio 3.57)
methyl methanesulfonate (Fig.1D, median RA 22.730 vs 8.330 in control, ratio 2.72)
UV radiation (Fig. 1E, median RA 44.970 vs 39.890 in control, ratio 1.125).

Also the efficiency of DNA repair from RA patients after the introduction both bleomycin or TBH affected is affected by the basal state and the cell death induced by drug treatment and aphidicolin. The results of this experiment should be evaluated after the analysis of cell death.

In the discussion the authors affirm that "Elevated sensitivity...to DNA damaging agents that trigger BER suggest that one key component of BER is affected and an additional study...”. "This probably explain higher level of DNA strand breaks observed in RA patients as compare with healthy subjects by Us and others [2,3,9] but it does not explain impaired repair of oxidative and DSBs repair".
1) On what basis do the authors claim it is a defect in the BERs? They themselves claim that further experiments are necessary to confirm the hypothesis (lines 143-144).
2) about "...impaired repair of oxidative..", the authors do not consider that high levels of oxidative stress can depend on various causes (NADPH oxidase, metabolism, mitochondria), and that a stressed sistem is more sensible to damage and repairs more slowly, which can induce cell death (10.1038/s12276-020-0384-2).
Again, aphidicolin inhibits both BER (https://doi.org/10.1038/sj.onc.1205561) and NER (https://doi.org/10.1093/mutage/gep039) enzymes; without considering that it inhibits DNA replication causing further damage in a already stressed sistem.

Minor issues:
The authors should indicate drug concentrations and treatment time.

Author Response

Response to Reviewer 3 Comments

Point 1: Figures 1B-1D show the percent of damage induced by TBH, bleomycin, methyl methanesulfonate and UV radiation between RA patients and control. How can the authors state that the observed differences are significant considering baseline differences between the two groups?

Response 1: Agreed, we must consider baseline differences between the two studied groups! We did it during DNA repair analysis but we did not it during DNA damage analysis. Therefore, we recalculated all data/graphs regarding DNA damage analysis taking into account the differences between the damage resulting from exposure to the DNA damaging compounds and basal, endogenous damage level. We have still observed the statistical significance in the level of PBMC sensitivity isolated from RA patients vs control group after exposure to: TBH, bleomycin, MMS (see novel graphs in Figure 1B’,1C’ and 1D’). However, we did not observed after baseline correction difference in the sensitivity of PMBC to UV radiation. These changes was inserted into manuscript text and figure 1. In the manuscript, we changed the fragment "We also show difference between level of DNA damage between PBMC exposed to UV radiation isolated from RA patients and compare to control (Fig. 1E) (median RA- 44.970, 25% = 30.650, 61.050 vs 39.890, 25% = 24.255, 75% = 56.220 at control), however it was the lowest and the discrepancy was less than 5 percent." into „However, we didn't notice the difference between level of DNA damage between PBMC exposed to UV radiation isolated from RA patients and compare to control (Fig. 1E) (median RA- 44.970, 25% = 30.650, 61.050 vs 39.890, 25% = 24.255, 75% = 56.220 at control).

Point 2: Have the authors measured drug treatment-induced cell death levels?

Response 2: Yes. We always have done it before DNA damage and repair studies, to determine the appropriate concentration of DNA damaging agents. It is good practice to have at least 80% of living cells. All drugs in the concentrations used for study fulfilled this requirement as we seen in XTT assay. We have added this information into the manuscript.

Point 3: Have the authors considered that the ratio between the two groups tends to drop after the various treatments? See baseline levels(Fig.1A, median RA 9.280 vs 2.090 in control, ratio 4.44) TBH (Fig.1B, median RA 22.330 vs 9.325 in control, ratio 2.39) ,  bleomycin (Fig.1C, median RA 44.190 vs 12.370 in control, ratio 3.57),  methyl methanesulfonate (Fig.1D, median RA 22.730 vs 8.330 in control, ratio 2.72), UV radiation (Fig. 1E, median RA 44.970 vs 39.890 in control, ratio 1.125).

Response 3: Yes, however this ratio was statistically important even after baseline correction (excluding UV). We obtained a very similar ratios as compared to values without baseline correction and they are respectively: for TBH (median RA 13.28 vs 6.62 in control, ratio 2.0), bleomycin TBH (median RA 34.65 vs 9.89 in control, ratio 3.5), methyl methanesulfonate (median RA 13.67 vs 5.9 in control, ratio 2.31) and UV radiation (median RA 36.1 vs 37.6 in control, ratio 0.96).

Point 4: Also the efficiency of DNA repair from RA patients after the introduction both bleomycin or TBH affected is affected by the basal state and the cell death induced by drug treatment and aphidicolin. The results of this experiment should be evaluated after the analysis of cell death.

Response 4: The effectiveness of DNA repair was estimated based on the percentage decrease in DNA damage, not quantitative. The repair curves was determined for both RA patient and healthy subjects, based on the time intervals in which DNA damage was measured. Each of the quantitative measurements obtained at a given time was converted into a percentage. We treated the first measurement at time 0 as 100% DNA damage. In this case, the quantitative difference resulting from basal lesions between the examined groups is not significant, because for both the RA group and healthy people the level of initial damage is 100%. Then, based on subsequent time points, the percentage of damage caused by subsequent measurements was calculated. The final analysis was to compare the obtained curves using the ROC curve. In addition, as we pointed out in the article, the analysis of the measurement effectiveness was made only for damage resulting from exposure to TBH and bleomycin. Aphidicolin was only used to measure the sensitivity of PBMCs to DNA damage resulting from exposure to MMS. In no other case was aphidicoline used.

Point 5: On what basis do the authors claim it is a defect in the BERs? They themselves claim that further experiments are necessary to confirm the hypothesis (lines 143-144).

Response 5: We only suggest that BER is affected in RA. As You noticed we realize that further experiments are necessary to confirm the hypothesis. However, our data suggest it - we used to estimate the level of damage and the effectiveness of PBMC repair tert-butyl hydroperoxide (TBH; T-BOOH). TBH induces oxidative stress in eukaryotic cells (doi.10.1155/2013/152909). TBH significantly increased the amount of oxidative base - 8-oxo-dG. 8-oxo-dG is the main substrate for glycosylases involved in BER. Therefore, the increased level of PBMC sensitivity to damage induced by TBH and less effective repair relative to the control group may suggest a lower efficiency of the BER system in RA patients than healthy subjects. However, the additional studies using specific BER substrates are needed. We add this information into discussion section.

Point 6: about "...impaired repair of oxidative..", the authors do not consider that high levels of oxidative stress can depend on various causes (NADPH oxidase, metabolism, mitochondria), and that a stressed system is more sensible to damage and repairs more slowly, which can induce cell death (10.1038/s12276-020-0384-2).

Response 6: The statement "impaired repair of oxidative" was a mental abbreviation due to the fact that the form of the article is „short” and therefore we did not want to write down to fit in the allowed number of words. We are also well aware that disturbed oxidative repair can be caused by many factors, including the accumulation of reactive oxygen species and their direct and indirect interaction in the cell, insufficient antioxidants and what you mentioned, NADPH oxidase, so it is important to know the cause of the disorder of oxidative repair. Anyway, we have added an explanation in the manuscript.

Point 7: Again, aphidicolin inhibits both BER (https://doi.org/10.1038/sj.onc.1205561) and NER (https://doi.org/10.1093/mutage/gep039) enzymes; without considering that it inhibits DNA replication causing further damage in an already stressed system.

Response 7: Exactly! Aphidicolin inhibits both BER as well as NER, however in our study aphidicolin is not a direct compound that induces DNA damage. Aphidicolin only increased the efficiency of MMS.  MMS is a direct acting methylating agent which produces apurinic sites that are transformed into DNA single-strand breaks by BER. Single-strand breaks are visible in comet assay. 10.1096/fj.201900308R 10.1074/jbc.M306592200 10.1016/s0027-5107(99)00034-2 PMID: 10197627 10.1016/s0921-8777(98)00053-6 10.1007/s002940050313 PMID: 8631315. We have added this information into the manuscript.

Point 8: The authors should indicate drug concentrations and treatment time.

Response 8: We have done so.

With many thanks for valuable comments,

Round 2

Reviewer 1 Report

Well done.

Reviewer 3 Report

The article has improved, however I believe that the DNA repair efficiency is intrinsically different between RA patients and controls, this is because oxidative stress in RA patients is very high (it remains high about 3 days in cultured cells), see:10.1371/journal.pone.0014003, 10.1126/scisignal.2004592.
I would recommend studying clonal selection in a future paper: 10.3390/biom10030446.